# Development of new real-time PCR assays for detection and species differentiation of *Plasmodium ovale*

Wenqiao He[1,2], Rachel Sendor[3], Varun R. Potlapalli[1], Melchior M. Kashamuka[4], Antoinette K. Tshefu[4], Fernandine Phanzu[5], Albert Kalonji[5], Billy Ngasala[6], Kyaw Lay Thwai[1], Jonathan J. Juliano[1,3], Jessica T. Lin[1], Jonathan B. Parr[1] *

1 Division of Infectious Diseases and Institute for Global Health and Infectious Diseases, University of North Carolina at Chapel Hill, Chapel Hill, North Carolina, United States of America, 2 Department of Epidemiology, School of Public Health, Guangdong Provincial Key Laboratory of Tropical Disease Research, Southern Medical University, Guangzhou, China, 3 Department of Epidemiology, Gillings School of Global Public Health, University of North Carolina at Chapel Hill, Chapel Hill, North Carolina, United States of America, 4 Kinshasa School of Public Health, Kinshasa, Democratic Republic of the Congo, 5 SANRU Asbl, Kinshasa, Democratic Republic of the Congo, 6 Muhimbili University of Health and Allied Sciences, Dar Es Salaam, Tanzania

* jonathan_parr@med.unc.edu

**Data Availability Statement:** All data and analysis R code underlying reported findings have been

## Abstract

### Background

The parasite species Plasmodium ovalecurtisi (P. ovalecurtisi) and Plasmodium ovalewallikeri (P. ovalewallikeri), formerly known as Plasmodium ovale, are endemic across multiple African countries. These species are thought to differ in clinical symptomatology and latency, but only a small number of existing diagnostic assays can detect and distinguish them. In this study, we sought to develop new assays for the detection and differentiation of *P. ovalecurtisi* and *P. ovalewallikeri* by leveraging recently published whole-genome sequences for both species.

### Methods

Repetitive sequence motifs were identified in available *P. ovalecurtisi* and *P. ovalewallikeri* genomes and used for assay development and validation. We evaluated the analytical sensitivity of the best-performing singleplex and duplex assays using synthetic plasmids. We then evaluated the specificity of the duplex assay using a panel of samples from Tanzania and the Democratic Republic of the Congo (DRC), and validated its performance using 55 *P. ovale* samples and 40 non-ovale *Plasmodium* samples from the DRC.

### Results

The best-performing *P. ovalecurtisi* and *P. ovalewallikeri* targets had 9 and 8 copies within the reference genomes, respectively. The *P. ovalecurtisi* assay had high sensitivity with a 95% confidence lower limit of detection (LOD) of 3.6 parasite genome equivalents/μl, while the *P. ovalewallikeri* assay had a 95% confidence LOD of 25.9 parasite genome

provided as part of the submitted article and
https://github.com/Wenqiao33/P.ovale_assays.

**Funding:** This study was funded by the US National Institutes of Health (NIH R21AI148579 to JBP and JTL). It was partly supported by the Global Fund to Fight AIDS, Tuberculosis, and Malaria (MK, AT, FP, AK; DRC sample collection); NIH R01AI137395 (JTL and BN; Tanzania sample collection), K24AI134990 (JJJ), and T32AI070114 (RS). The funders had no role in study design, data collection and analysis, decision to publish, or preparation of the manuscript.

**Competing interests:** I have read the journal's policy and the authors of this manuscript have the following competing interests: JBP reports research support from Gilead Sciences, non-financial support from Abbott Laboratories, and consulting for Zymeron Corporation, all outside the scope of the manuscript. All other authors declare no competing interests.

equivalents/µl. A duplex assay targeting both species had 100% specificity and 95% confidence LOD of 4.2 and 41.2 parasite genome equivalents/µl for *P. ovalecurtisi* and *P. ovalewallikeri*, respectively.

## Conclusions

We identified promising multi-copy targets for molecular detection and differentiation of *P. ovalecurtisi* and *P. ovalewallikeri* and used them to develop real-time PCR assays. The best performing *P. ovalecurtisi* assay performed well in singleplex and duplex formats, while the *P. ovalewallikeri* assay did not reliably detect low-density infections in either format. These assays have potential use for high-throughput identification of *P. ovalecurtisi*, or for identification of higher density *P. ovalecurtisi* or *P. ovalewallikeri* infections that are amenable to downstream next-generation sequencing.

### Author summary

Non-falciparum malaria appears to be on the rise, especially in settings where *P. falciparum* transmission is declining. *Plasmodium ovalecurtisi* and *Plasmodium ovalewallikeri* are neglected parasites that can cause relapsing malaria and are thought to differ in clinical symptomatology and latency. However, few existing diagnostic assays can detect and distinguish them. Most target the 18S rRNA gene of both *P. ovalecurtisi* and *P. ovalewallikeri* with potential for cross-reactivity at higher parasite densities, and are not well-suited for high-throughput use, hindering our understanding of their epidemiology. Mining recently available *P. ovalecurtisi* and *P. ovalewallikeri* reference genomes, we identify new multi-copy targets for molecular detection and develop novel singleplex and duplex real-time PCR assays capable of species differentiation. These assays are highly specific and require short turn-around time. The *P. ovalecurtisi* assay performed well, while the *P. ovalewallikeri* assay did not reliably detect low-density infections. These assays provide new options for high-throughput studies of *P. ovalecurtisi* infection, as well as identification of higher density infections amenable to next-generation sequencing of both species.

## Introduction

Malaria remains a major global health concern despite decades of sustained investment in elimination efforts. Though most malaria control programs within Africa prioritize *Plasmodium falciparum*, the parasite species responsible for most deaths, increasing evidence confirms co-circulation of other neglected *Plasmodium* species that cause human malaria [1–4]. Recent surveys reveal a previously unappreciated burden of *Plasmodium ovalecurtisi* and *Plasmodium ovalewallikeri* in multiple African countries [5,6], where relapsing malaria caused by these parasites may prove to be an obstacle to malaria elimination efforts [7,8]. *P. ovalecurtisi* and *P. ovalewallikeri* (previously known as *P. ovale curtisi* and *P. ovale wallikeri*), which were formerly known as *Plasmodium ovale* [9–12], have potential differences in clinical symptomatology and latency [13], but few existing diagnostic assays have ability to detect and distinguish them. Some require separate PCR runs, multiple steps (nested assays, agarose gel electrophoresis, and/or sequencing), or prolonged cycling time that increases risk of false-positive results [14–19].

Differentiation of *P. ovalecurtisi* and *P. ovalewallikeri* is not currently possible using microscopy, the gold standard for malaria diagnosis in the field [20]. *P. ovalecurtisi* and *P. ovalewallikeri* infections often occur as mixed infections at low density, and are morphologically indistinguishable on blood slides [21,22]. Furthermore, widely used malaria rapid diagnostic tests (RDTs) fail to detect samples with low parasite densities and cannot distinguish parasite species other than *P. falciparum* and *Plasmodium vivax* [23,24]. Thus, alternative methods are required to identify these neglected species.

Molecular methods (S1 Table) [14–19,25–28] are more sensitive and specific for *P. ovale* detection than microscopic examination or RDTs, but most existing assays target the 18S rRNA gene of both *P. ovalecurtisi* and *P. ovalewallikeri*, leading to potential cross-reactivity and a lack of complete species specificity. A duplex real-time PCR assay targeting the reticulocyte-binding protein homologue (*porbp2*) gene for *P. ovalecurtisi* and *P. ovalewallikeri* detection was published in 2011; however, results of the melt-curve analysis can be hard to interpret [15]. A nested PCR assay developed in 2013 targets the tryptophan-rich antigen (*potra*) gene and can detect samples with 2–10 parasites/µl [16], but this assay requires multiple steps (nested assay, agarose gel electrophoresis, and sequencing) and long turnaround time. Nested PCR targeting the *Plasmodium* mitochondrial cytochrome c oxidase III (*cox3*) gene can also differentiate species, but it requires agarose gel electrophoresis and sequencing [18]. Available single-target quantitative real-time PCR assays require separate runs to distinguish *P. ovalecurtisi* and *P. ovalewallikeri* [14,17,19].

Because of the limitations of the existing assays, most studies have not distinguished *P. ovalecurtisi* and *P. ovalewallikeri* [29]. However, recently released *P. ovalecurtisi* and *P. ovalewallikeri* genomes (PocGH01 and PowCR01) provide opportunities for improved molecular assay development [30]. To improve our understanding of the epidemiology of *P. ovalecurtisi* and *P. ovalewallikeri* malaria, we mined publicly available *P. ovalecurtisi* and *P. ovalewallikeri* genomes to identify novel multi-copy targets and developed new qualitative real-time PCR assays. Our new assays have high specificity and can be duplexed. Performance of the *P. ovalecurtisi* assay was superior to the *P. ovalewallikeri* assay, limiting the duplex assay's ability to investigate their relative prevalence. These assays offer new options for high-throughput *P. ovalecurtisi* epidemiological analyses and identification of *P. ovalecurtisi* and *P. ovalewallikeri* samples amenable to downstream next-generation sequencing.

## Materials and Methods

### Ethics statement

Existing samples from previous studies were chosen based on convenience. DRC samples were collected as part of a 2017 study investigating malaria diagnostic test performance in three provinces, Kinshasa, Bas-Uele, and Sud-Kivu [31]. Tanzania samples were collected from participants enrolled in a malaria transmission study in rural Bagamoyo district from 2018–2019 [14,29,32]. Enrolled subjects provided written informed consent or assent; for children, written parental consent was obtained. Ethical approvals for these studies were obtained from the Kinshasa School of Public Health (ESP/CE/07B/2017), Muhimbili University of Health and Allied Sciences (MUHAS/DA.282/298/01/C), and the University of North Carolina at Chapel Hill (IRB#: 17–0155).

### Mining and selection of multi-copy targets in *P. ovalecurtisi* and *P. ovalewallikeri* genomes

Using the publicly available *P. ovalecurtisi* (PocGH01) and *P. ovalewallikeri* (PowCR01) reference genomes obtained from the NIH National Center for Biotechnology Information (NCBI) database, we identified sequence motifs of 100 base-pairs (bp) in length with $\geq 6$ copies using

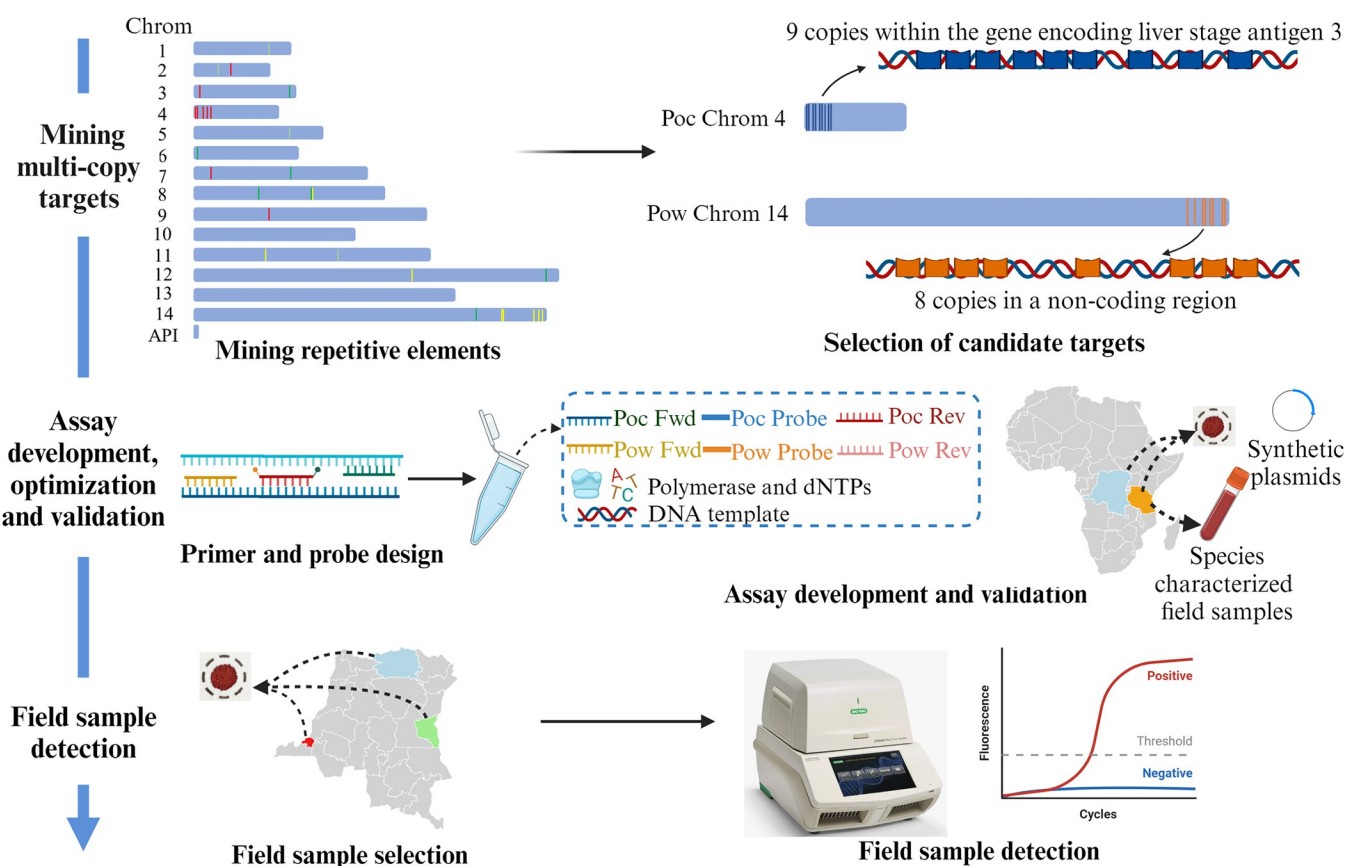

**Fig 1. Approach to develop real-time PCR assays for the detection and differentiation of *P. ovalecurtisi* and *P. ovalewallikeri*.** Figure created using Biorender.com. Maps of Africa and the DRC were created using R software. Abbreviations: Poc = *P. ovalecurtisi*; Pow = *P. ovalewallikeri*.

*Jellyfish* (version 2.2.10) [33] (Fig 1). Sequences with low GC content (< 25%) and highly repetitive short sequences were excluded. The remaining multi-copy targets were aligned to NCBI nt database using *blastn* to investigate their specificity. Sequences aligned to other *Plasmodium* parasites were excluded. We then re-aligned the remaining targets to the *P. ovalecurtisi* and *P. ovalewallikeri* genomes separately using *blastn* to investigate their copy numbers in each genome. Candidate diagnostic assay targets for *P. ovalecurtisi* and *P. ovalewallikeri* were selected based on species-specificity and copy numbers. Primer and probe sets were designed manually using Oligo Calc [34] and DNAMAN (version 9, Lynnon BioSoft, Quebec City, Canada) to estimate primer and probe melting temperatures and to avoid self-complementarity and primer dimers (S2 Table).

## Assay development and optimization

A panel of 15 well-characterized *P. ovalecurtisi* and *P. ovalewallikeri* field samples and six non-ovale *Plasmodium* laboratory controls were selected for assay development and analytical specificity analysis. Field samples included 11 *P. ovalecurtisi* and four *P. ovalewallikeri* leukodepleted blood samples and dried blood spot (DBS) samples from Tanzania and the Democratic Republic of the Congo (DRC); species identification was conducted using the published assays [14]. Laboratory controls included two *P. falciparum*, one *P. malariae*, two *P. vivax*, and one *P. knowlesi* dried blood spot samples from an external quality assurance program [35]. DNA from dried blood spot (DBS) samples, each spot containing approximately 70μl whole blood,

was extracted using Chelex 100 (Bio-Rad, Fishers, Indiana, USA) and eluted into 150μl final volume [36]. DNA from leukodepleted blood samples was extracted using the QIAamp DNA Mini Kit (Qiagen, Mettmann, North Rhine-Westphalia, Germany) according to manufacture instructions. Parasite densities were estimated using a semi-quantitative real-time PCR assay targeting the 18S rRNA gene of both *P. ovalecurtisi* and *P. ovalewallikeri* as previously described [5]. The *P. ovalecurtisi* versus *P. ovalewallikeri* species was determined using published assays as the gold standard [14].

Primer sets with the best specificity for *P. ovalecurtisi* and *P. ovalewallikeri* versus this panel of samples were selected for further development. Singleplex assays for *P. ovalecurtisi* and *P. ovalewallikeri* detection were optimized using synthetic plasmids (Azenta Life Sciences, Indianapolis, Indiana, USA) containing targets (S3 Table) for *P. ovalecurtisi* and *P. ovalewallikeri* detection. A range of annealing temperatures and primer and probe concentrations were tested to identify the optimal reaction conditions. Finally, a duplex qualitative real-time PCR assay that combined the singleplex assays was developed, in order to detect and differentiate *P. ovalecurtisi* and *P. ovalewallikeri* in a single reaction tube. Duplex assay optimization was performed using synthetic plasmids described above. Optimal reaction conditions were determined by testing a range of annealing temperatures and of primer and probe concentrations.

All reactions were performed using a CFX384 Touch Real-Time PCR Detection System (Bio-Rad, Hercules, CA). All optimization analyses were performed in duplicate. Non-template controls (nuclease-free water) and serially diluted *P. ovalecurtisi* and *P. ovalewallikeri* plasmid DNA solutions were included in each real-time PCR run.

## Analytical sensitivity and specificity

We determined the analytical sensitivity of the best performing singleplex and duplex assays using serially diluted plasmid DNA. A total of 129 *P. ovalecurtisi* and 186 *P. ovalewallikeri* plasmid DNA replicates were tested to determine the analytical sensitivity of the singleplex assays (S4 Table). For the duplex assay, a total of 104 *P. ovalecurtisi* and 161 *P. ovalewallikeri* plasmid replicates were used (S5 Table). Probit analysis was used to estimate the 95% confidence lower limits of detection [37]. We then determined the analytical specificity of the duplex assay using the same panel of 15 well-characterized *P. ovalecurtisi* and *P. ovalewallikeri* field samples [14] and six non-ovale *Plasmodium* laboratory controls in duplicate as described above.

## Validation using field samples

The duplex assay's clinical sensitivity and specificity were assessed using 95 dried blood spot samples selected from a large sample set from a previous study conducted in the DRC [31], including 55 *P. ovalecurtisi* and/or *P. ovalewallikeri* samples identified using published PCR assay [5], and 40 non-ovale *Plasmodium* samples (20 *P. falciparum* infections, 10 *P. malariae* infections, and 10 *P. falciparum* and *P. malariae* mixed infections) [31]. DNA was extracted from DBS using Chelex 100 as described above. *Plasmodium* species and parasite densities were identified using real-time PCR assays for both *P. ovalecurtisi* and *P. ovalewallikeri*, *P. falciparum*, and *P. malariae* as previously described [5,38,39], with samples positive in duplicate selected for use during validation of the present assay. Results of the previously published singleplex 18S rRNA real-time PCR assay for both *P. ovalecurtisi* and *P. ovalewallikeri* was used as the gold standard for clinical sensitivity and specificity calculations [5].

## Statistical analysis

Statistical analysis was performed using R software (version 4.2.0; R Core Team, Vienna, Austria) in RStudio (version 2022.02.2). Maps and figures were generated using the *ggplot2*

**Table 1. Best performing primers and probes for *P. ovalecurtisi* and *P. ovalewallikeri* detection.**

| Name | Sequence (5'→3') |
| --- | --- |
| Poc_Fwd | GTTRCCAAATATGCTATCACTTAC |
| Poc_Rev | GTARCACAAAACGACGAGAC |
| Poc_Probe | FAM—TACATCTTCTTCAAAGTTGYCATAYGCAT—BHQ1 |
| Pow_Fwd | GRRTCTTCTGAACTTTGRAATG |
| Pow_Rev | CATCAAGGRTATCCATTTCA |
| Pow_Probe | VIC—AACAAYCACTTCAACATCAA—BHQ1 |

(version 3.4.4), *sf* (version 1.0.16), *rnaturalearth* (version 1.0.1), and *rnaturalearthdata* (version 1.0.0) packages, and the study schematic was generated using BioRender. Spatial data were downloaded from the Database of Global Administrative Areas (GADM) [40].

## Results

### *P. ovalecurtisi* and *P. ovalewallikeri* target selection and assay development

A total of 2,585 and 3,978 sequences of 100 bp in length with ≥6 repeats were found in the *P. ovalecurtisi* and *P. ovalewallikeri* reference genomes, respectively. Targets with low GC content, highly repetitive short sequences, or aligned to other *Plasmodium* parasite genomes were excluded. A total of three potential assay targets with ≥8 copies in each of the *P. ovalecurtisi* and *P. ovalewallikeri* genomes were selected. Focusing on these potential targets, we designed five and three primer and probe sets for *P. ovalecurtisi* and *P. ovalewallikeri*, respectively (S2 Table). After testing all primer and probe sets using a panel of 15 well-characterized *P. ovalecurtisi* and *P. ovalewallikeri* field samples and six laboratory non-ovale *Plasmodium* controls, we selected two primer and probe sets with the best specificity for *P. ovalecurtisi* and *P. ovalewallikeri*, respectively, for additional laboratory testing (Table 1). The selected *P. ovalecurtisi* target had nine copies within putative liver stage antigen 3 (*lsa3*) gene on chromosome 4 (LT594585.1: 9,968–11,125), while the *P. ovalewallikeri* target had eight copies in a non-coding region on chromosome 14 (LT594518.1: 1,842,975–1,844,586). Short distances (< 50 bp) were noted between the repetitive *P. ovalecurtisi* target motifs as well as between *P. ovalewallikeri* target motifs.

### Singleplex real-time PCR assay development

Using the primer sets and the corresponding probes with the best specificity for *P. ovalecurtisi* and *P. ovalewallikeri*, we developed singleplex assays for detection of each species. The optimized assay for *P. ovalecurtisi* was performed in a small volume of 10μl, including 7μl of reaction master-mix containing 2x FastStart Universal Probe Master (Rox) (Roche, Basel, Switzerland), primers and probes (240 nM of Poc_Fwd, 240 nM of Poc_Rev, 60 nM of Poc_Probe), and 3μl of DNA template (derived from approximately 1.4μl whole blood). Optimal thermocycling conditions were 2 min at 50˚C, 10 min at 95˚C, followed by 40 cycles of 15 s at 95˚C and 60 s at 58˚C. The optimized assay for *P. ovalewallikeri* was also performed in a small volume of 10μl, including 7μl of reaction master-mix containing 2x FastStart Universal Probe Master (Rox), primers and probes (300 nM of Pow_Fwd, 300 nM of Pow_Rev, 200nM of Pow_Probe), and 3μl of DNA template. The optimal thermocycling conditions were 2 min at 50˚C, 10 min at 95˚C, followed by 45 cycles of 15 s at 95˚C and 60 s at 56˚C. Samples with Ct values lower than 40 and 45 were called positive for *P. ovalecurtisi* and *P. ovalewallikeri*, respectively.

## Duplex real-time PCR assay development

Combining the singleplex assays, we optimized a duplex, qualitative real-time PCR assay for simultaneous detection and differentiation of *P. ovalecurtisi* and *P. ovalewallikeri* in a single reaction tube. The final, optimized duplex assay was performed in a small final volume of 10μl, including 7μl of reaction master-mix containing 2x FastStart Universal Probe Master (Rox) (Roche, Basel, Switzerland), primers and probes (240 nM of Poc_Fwd, 240 nM of Poc_Rev, 60 nM of Poc_Probe, 800 nM of Pow_Fwd, 800 nM of Pow_Rev, 320nM of Pow_Probe), and 3μl of DNA template. Optimal thermocycling conditions were 2 min at 50˚C, 10 min at 95˚C, followed by 45 cycles of 15 s at 95˚C and 60 s at 58˚C, allowing for detection of parasite DNA in less than two hours. Samples with Ct values lower than 45 for either species were called positive.

## Analytical sensitivity and specificity

The 95% confidence lower limits of detection of the singleplex *P. ovalecurtisi* and *P. ovalewallikeri* assays were 3.6 and 25.9 parasite genome equivalents/μl DNA template, respectively (S1 Fig and S4 Table). The 95% confidence lower limits of detection of the duplex assay were similar to the singleplex assays, at 4.2 and 41.2 parasite genome equivalents/μl DNA template, respectively (Figs 2A, S2 and S5 Table). All well-characterized *P. ovalecurtisi* and *P. ovalewallikeri* field samples were successfully detected and differentiated with no cross-reactivity between species, and no cross reactivity was found when the assay was applied to six non-ovale *Plasmodium* controls (Fig 2B).

## Validation using field samples

The duplex assay demonstrated perfect specificity for *P. ovalecurtisi* and *P. ovalewallikeri* and high sensitivity for *P. ovalecurtisi* when applied to 95 field samples collected in the DRC.

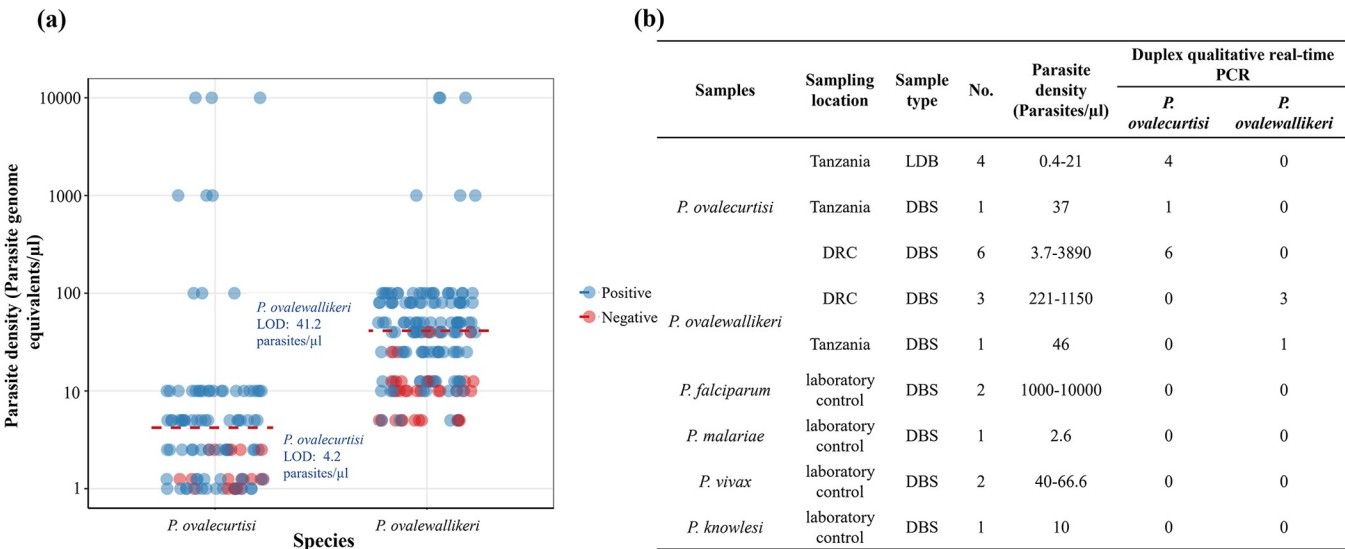

**(a)**

**(b)**

| Samples | Sampling location | Sample type | No. | Parasite density (Parasites/μl) | Duplex qualitative real-time PCR | |
|---|---|---|---|---|---|---|
| | | | | | *P. ovalecurtisi* | *P. ovalewallikeri* |
| *P. ovalecurtisi* | Tanzania | LDB | 4 | 0.4-21 | 4 | 0 |
| | Tanzania | DBS | 1 | 37 | 1 | 0 |
| | DRC | DBS | 6 | 3.7-3890 | 6 | 0 |
| *P. ovalewallikeri* | DRC | DBS | 3 | 221-1150 | 0 | 3 |
| | Tanzania | DBS | 1 | 46 | 0 | 1 |
| *P. falciparum* | laboratory control | DBS | 2 | 1000-10000 | 0 | 0 |
| *P. malariae* | laboratory control | DBS | 1 | 2.6 | 0 | 0 |
| *P. vivax* | laboratory control | DBS | 2 | 40-66.6 | 0 | 0 |
| *P. knowlesi* | laboratory control | DBS | 1 | 10 | 0 | 0 |

**Fig 2. Duplex *P. ovalecurtisi* and *P. ovalewallikeri* assay performance. A)** Analytical sensitivity when applied to multiple replicates of serially diluted plasmid DNA (n = 104 and 161 total replicates for *P. ovalecurtisi* and *P. ovalewallikeri*, respectively). Points are colored to display target detection (blue) versus no detection (red). The 95% lower limit of detection (LOD) determined using probit analysis is shown for each species. **B)** Analytical specificity versus genomic DNA extracted from a panel of well-characterized leukodepleted blood (LDB) and dried blood spot (DBS) samples from Tanzania and the DRC with *P. ovalecurtisi* and *P. ovalewallikeri* confirmed by published real-time PCR assays, and non-ovale *Plasmodium* samples from an external quality assurance program. All *P. ovalecurtisi* and *P. ovalewallikeri* samples were correctly identified, and no false-positives were observed among other *Plasmodium* species.

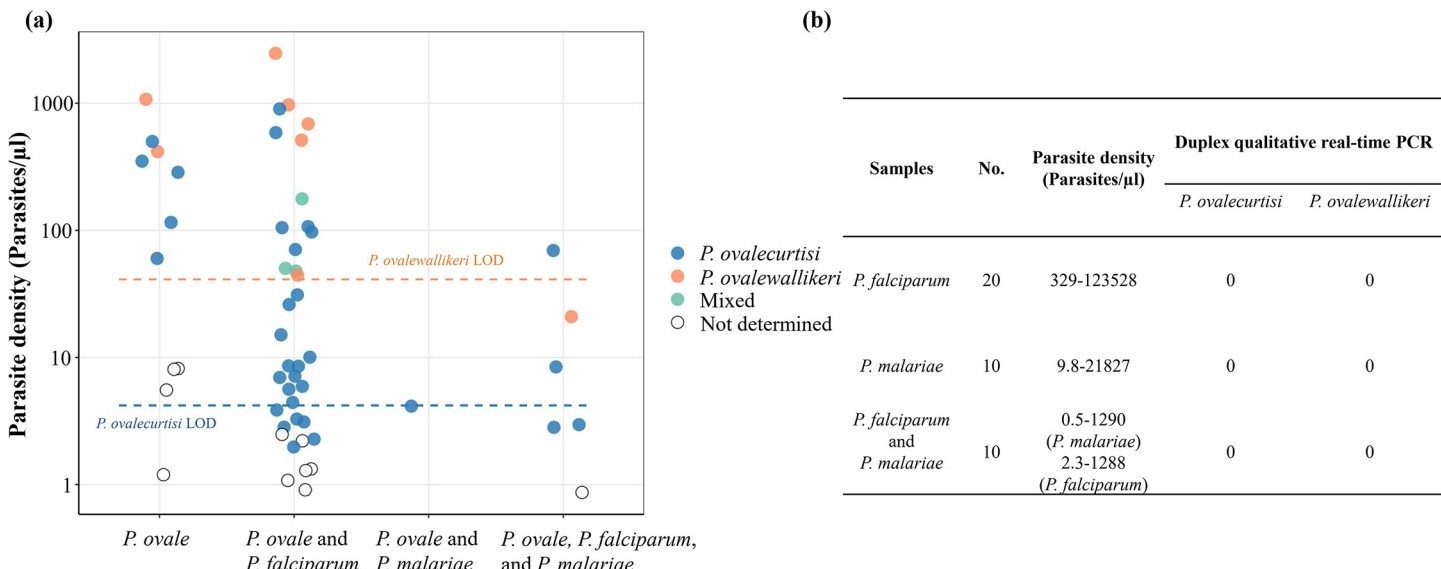

**Fig 3. Assay validation using field samples collected in the DRC.** Gold standard species identification was performed previously using a series of semi-quantitative real-time PCR assays targeting pan-*Plasmodium* 18S rRNA, followed by singleplex species-specific assays. **A)** Detection of known *P. ovale* PCR-positive samples with varying parasite densities and co-infection status. Analytical 95% lower limits of detection (LOD) are represented by dashed lines. **B)** No detection of other *Plasmodium* species across a range of parasite densities. Abbreviations: *P. ovale* = *P. ovalecurtisi* and/or *P. ovalewallikeri*.

Parasite densities of 55 *P. ovale*-positive field samples included in this study ranged from 0.9 to 2,468 parasites/μl DNA template; 29 (52.7%) samples had parasite densities <10 parasites/μl. The assay's overall sensitivity was 80%, successfully determining *P. ovale* species in 44 of the *P. ovale*-positive field samples (Fig 3A). False-negatives were limited to low-concentration samples, with 100% assay sensitivity for infections with >10 parasites/μl DNA template. The lowest parasite densities in which species could be determined were 2.0 and 20.9 parasites/μl DNA template for *P. ovalecurtisi* and *P. ovalewallikeri*, respectively. None of the 40 non-ovale *Plasmodium* field samples were detected by the duplex assay, consistent with 100% specificity (Fig 3B).

## Discussion

We mined recently published genomes of *P. ovalecurtisi* and *P. ovalewallikeri* to develop new real-time PCR assays that can be used to improve our understanding of their epidemiology in malaria-endemic countries. Recent studies have revealed a previously unappreciated burden of *P. ovalecurtisi* and *P. ovalewallikeri* in Africa [3, 5, 15, 29]. Though *P. ovalecurtisi* and *P. ovalewallikeri* are distinct species, only a small number of existing assays can distinguish them. Many are not well-suited to large studies, requiring separate assays for each species, multiple steps (nested assays, agarose gel electrophoresis, and/or sequencing), higher input volumes of DNA solution, and long turnaround time, with potential for cross-reactivity at higher parasite densities [14–19]. Because of the limits of the existing assays, most field studies do not distinguish *P. ovalecurtisi* and *P. ovalewallikeri*, and their prevalence and clinical features remain understudied [41–43].

Our assays are highly specific for *P. ovalecurtisi* and *P. ovalewallikeri*, but we observed differences in sensitivity for detection of *P. ovalecurtisi* and *P. ovalewallikeri*. This difference in sensitivity limits the duplex assay's use for studies of their relative prevalence. However, the *P. ovalecurtisi* singleplex or duplex assay is well-suited for high-throughput studies of

symptomatic *P. ovalecurtisi* infection. In contrast, the *P. ovalewallikeri* assay is well-suited to identify higher-density infections that are amenable to next-generation sequencing, but other more sensitive assays should be used for epidemiological analyses because our assay does not reliably detect lower-density infections. Thus, choice of assay and format should be informed by the user's specific objectives.

Our assay targets are distinct from those used in prior assays and take advantage of 100 bp repetitive motifs in the putative *lsa3* gene on *P. ovalecurtisi* chromosome 4 and a non-coding region on *P. ovalewallikeri* chromosome 14, respectively. Studies of *P. falciparum lsa3* indicate that it is an essential gene that encodes an antigen with tetrapeptide repeats of unclear function during the liver stage of infection [44–46]. Previous work confirmed conservation of *P. falciparum lsa3* in isolates collected from geographically diverse sites [45]. The non-coding *P. ovalewallikeri* repetitive motif we targeted has unclear function, with no obvious orthologues identified in publicly available databases. These targets appear to be conserved in the limited *P. ovalecurtisi* and *P. ovalewallikeri* genomes released to-date. We leveraged the repetitive nature of these poorly understood *P. ovalecurtisi* and *P. ovalewallikeri* targets to develop highly specific assays for *P. ovalecurtisi* and *P. ovalewallikeri*, and high sensitivity for *P. ovalecurtisi*.

Compared to published real-time PCR assays that mostly target *P. ovalecurtisi* and *P. ovalewallikeri* 18S rRNA genes [5, 17, 19], inclusion of distinct *P. ovalecurtisi* and *P. ovalewallikeri* targets enabled development of highly specific assays. The targets' copy numbers in our study are in the same range as those reported for 18S rRNA genes in *Plasmodium* genomes [47–49]. Similar limits of detection of *P. ovalecurtisi* were found between the published 18S rRNA PCR assay (1.5 parasites/μl) and our *P. ovalecurtisi* singleplex and duplex assays, while we observed inferior limits of detection for *P. ovalewallikeri* compared to some published assays (S1 Table). It is possible that the short distances between our *P. ovalecurtisi* targets and between *P. ovalewallikeri* targets decrease the PCR efficiency, offsetting sensitivity that might otherwise be achieved from their copy number.

We further evaluated the duplex assay using field samples from the DRC. Validation using field samples from the DRC confirmed robust species differentiation when the duplex assay was applied to *P. ovale* samples with >10 parasites/μl and 100% specificity across all parasite densities. Though its ability to identify *P. ovalewallikeri* in particular was limited at lower parasite densities, the simultaneous amplification of *P. ovalecurtisi* and *P. ovalewallikeri* DNA in a single reaction tube allows our assay to have shorter turnaround time and require less materials compared to published singleplex assays [17, 19]. The duplex assay had high specificity, high sensitivity for *P. ovalecurtisi* detection, short turnaround time, and capacity for high-throughput use.

Several limitations of our assays should be highlighted. First, the duplex assay's relatively low sensitivity at lower parasite densities, particularly for *P. ovalewallikeri* detection as noted above, limits its utility in epidemiological analyses and particularly among low-density or asymptomatic infections. This limitation could be overcome in the future by combining an 18S rRNA assay capable of detecting both *P. ovalecurtisi* and *P. ovalewallikeri* (e.g. such as that used by Mitchell et al. [5]) with our *P. ovalecurtisi lsa3* assay, allowing definitive identification of *P. ovalecurtisi* (18S rRNA assay-positive, *P. ovalecurtisi lsa3*-positive) and deductive identification of *P. ovalewallikeri* mono-infection (18S rRNA assay-positive, *P. ovalecurtisi lsa3*-negative). Second, the assays were optimized with high-throughput applications in mind, but lower-throughput approaches may be more appropriate in some cases. For example, users with smaller numbers of samples or willing to expend larger DNA volumes could consider increasing sample volumes to improve sensitivity. Careful validation of this approach within one's own lab is critical to ensure assay specificity is maintained. Third, our assays target two non-essential genomic regions at risk of deletion or disruption if future treatment choices are

tied to diagnosis, as has been proposed for *P. falciparum* and observed for *Chlamydia trachomatis* non-essential diagnostic targets [50, 51]. However, this hypothetical threat is unlikely to be realized any time soon. Malaria programs in Africa focus largely on *P. falciparum* and do not routinely offer radical cure to clear *P. ovalecurtisi* and *P. ovalewallikeri* hypnozoites. Finally, these assays were developed based on *P. ovalecurtisi* and *P. ovalewallikeri* genomes from Africa. More sequences from other regions are needed to assess for variation in the primer and probe targets.

In conclusion, we developed and validated novel, highly specific real-time PCR assays capable of detection and differentiation of *P. ovalecurtisi* and *P. ovalewallikeri*. Though its ability to identify *P. ovalewallikeri* was limited at lower parasite densities, the duplex assay's streamlined work-flow reduces complexity and may be suitable for specific use cases. We recommend these assays for high-throughput analyses of symptomatic *P. ovalecurtisi* malaria and for identification of higher-density *P. ovalecurtisi* or *P. ovalewallikeri* infections that may be amenable to sequencing. As some countries progress toward malaria elimination, improved assays for *P. ovalecurtisi* and *P. ovalewallikeri* like those presented here will become more important and open the way to improved understanding of *P. ovalecurtisi* and *P. ovalewallikeri* epidemiology and clinical impact, and ultimately inform elimination strategies.

## Supporting information

**S1 Table. Molecular assays to distinguish *P. ovalecurtisi* and *P. ovalewallikeri*.**
(DOCX)

**S2 Table. Candidate primer and probe sets evaluated for the detection of *P. ovalecurtisi* and *P. ovalewallikeri*.**
(DOCX)

**S3 Table. Sequences contained in synthetic plasmids to determine assay analytical sensitivity.**
(DOCX)

**S4 Table. Limits of detection of the optimized, singleplex *P. ovalecurtisi* and *P. ovalewallikeri* assays versus serially diluted plasmid DNA.** Parasite density = plasmid DNA copy number/copy number of the target in the parasite genome.
(DOCX)

**S5 Table. Limits of detection of the optimized, duplex *P. ovalecurtisi* and *P. ovalewallikeri* assay versus serially diluted plasmid DNA.** Parasite density = plasmid DNA copy number/copy number of the target in the parasite genome.
(DOCX)

**S1 Fig. 95% lower limits of detection for singleplex assays, determined using probit analysis.** A) *P. ovalecurtisi* singleplex assay 95% lower limit of detection (3.6 parasites/μl [95% CI 2.7–6]). B) *P. ovalewallikeri* singleplex assay 95% lower limit of detection (25.9 parasites/μl [95% CI 22–33.6]). Confidence intervals are shown in lighter shade.
(TIF)

**S2 Fig. 95% lower limits of detection for duplex assay, determined using probit analysis.** A) *P. ovalecurtisi* 95% lower limit of detection (4.2 parasites/μl [95% CI 3.1–9.5]). B) *P. ovalewallikeri* 95% lower limit of detection (41.2 parasites/μl [95% CI 33.3–58.3]). Confidence intervals are shown in lighter shade.
(TIF)

## Acknowledgments

We thank the study teams and participants in the DRC and Tanzania research studies from which samples were derived. The following reagents were obtained through BEI Resources, NIAID, NIH: diagnostic plasmid containing the small subunit ribosomal RNA gene (18S) from *Plasmodium ovale*, MRA-180, contributed by Peter A. Zimmerman.

## Author Contributions

**Conceptualization:** Wenqiao He, Jonathan B. Parr.

**Formal analysis:** Wenqiao He.

**Funding acquisition:** Rachel Sendor, Melchior M. Kashamuka, Antoinette K. Tshefu, Fernandine Phanzu, Albert Kalonji, Billy Ngasala, Jonathan J. Juliano, Jessica T. Lin, Jonathan B. Parr.

**Investigation:** Wenqiao He, Rachel Sendor, Varun R. Potlapalli, Kyaw Lay Thwai.

**Methodology:** Wenqiao He, Jonathan J. Juliano, Jessica T. Lin, Jonathan B. Parr.

**Project administration:** Jonathan B. Parr.

**Resources:** Melchior M. Kashamuka, Antoinette K. Tshefu, Fernandine Phanzu, Albert Kalonji, Billy Ngasala.

**Visualization:** Wenqiao He.

**Writing – original draft:** Wenqiao He.

**Writing – review & editing:** Wenqiao He, Rachel Sendor, Varun R. Potlapalli, Melchior M. Kashamuka, Antoinette K. Tshefu, Fernandine Phanzu, Albert Kalonji, Billy Ngasala, Kyaw Lay Thwai, Jonathan J. Juliano, Jessica T. Lin, Jonathan B. Parr.

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
