## [Decision Letter · Decision Letter 0]

6 Feb 2024

Dear Dr. Parr,

Thank you very much for submitting your manuscript "A novel duplex qualitative real-time PCR assay for the detection and differentiation of Plasmodium ovale curtisi and Plasmodium ovale wallikeri malaria" for consideration at PLOS Neglected Tropical Diseases. As with all papers reviewed by the journal, your manuscript was reviewed by members of the editorial board and by several independent reviewers. In light of the reviews (below this email), we would like to invite the resubmission of a significantly-revised version that takes into account the reviewers' comments. 

I concur with the comments of the three reviewers. The concerns of reviewers 1 and 3 with respect to the lower sensitivity for P. ovalewallikeri is particularly important: 40 P/µL is very close to the limit of detection by routine thick smear examination, and this is likely to lead to false negative results for many of the infections found in the field, especially in asymptomatic individuals. These two reviewers also suggest that the central claim that the assay represents an improvement on other current molecular assays is not justified. The significance of the epidemiological analyses has also been questioned by these two reviewers.

I suggest that significant major amendments are made to any revised manuscript in response to the detailed comments made by the reviewers.

We cannot make any decision about publication until we have seen the revised manuscript and your response to the reviewers' comments. Your revised manuscript is also likely to be sent to reviewers for further evaluation.

Sincerely,

Georges Snounou, Ph.D.

Guest Editor

Paul Mireji

Section Editor

I concur with the comments of the three reviewers. The concerns of reviewers 1 and 3 with respect to the lower sensitivity for P. ovalewallikeri is particularly important: 40 P/µL is very close to the limit of detection by routine thick smear examination, and this is likely to lead to false negative results for many of the infections found in the field, especially in asymptomatic individuals. These two reviewers also suggest that the central claim that the assay represents an improvement on other current molecular assays is not justified. The significance of the epidemiological analyses has also been questioned by these two reviewers.

I suggest that significant major amendments are made to any revised manuscript in response to the detailed comments made by the reviewers.

Reviewer's Responses to Questions

**Key Review Criteria Required for Acceptance?**

**Methods**

-Are the objectives of the study clearly articulated with a clear testable hypothesis stated?

-Is the study design appropriate to address the stated objectives?

-Is the population clearly described and appropriate for the hypothesis being tested?

-Is the sample size sufficient to ensure adequate power to address the hypothesis being tested?

-Were correct statistical analysis used to support conclusions?

-Are there concerns about ethical or regulatory requirements being met?

Reviewer #1: (No Response)

Reviewer #2: See attachment

Reviewer #3: The objectives and study design are appropriate. The sample size for the validation of the novel assay is appropriate, though limited to geographically restricted samples, while that for the epidemiological analyses is not adequate.

**Results**

-Does the analysis presented match the analysis plan?

-Are the results clearly and completely presented?

-Are the figures (Tables, Images) of sufficient quality for clarity?

Reviewer #1: (No Response)

Reviewer #2: See attachment

Reviewer #3: The results and analyses are clearly presented, as are the figures

**Conclusions**

-Are the conclusions supported by the data presented?

-Are the limitations of analysis clearly described?

-Do the authors discuss how these data can be helpful to advance our understanding of the topic under study?

-Is public health relevance addressed?

Reviewer #1: (No Response)

Reviewer #2: See attachment

Reviewer #3: The limitations listed by the authors are sufficient to weaken some of the conclusions presented and the relevance of the new assay to public health.

**Editorial and Data Presentation Modifications?**

Reviewer #1: (No Response)

Reviewer #2: See attachment

Reviewer #3: Minor comments

Line 73 Reference 1 is not adequate in this context.

Line 316/Line 322 LSA3 might be non-essential for blood-stage parasites, but it is essential for hepatic parasites. Furthermore, there is no evidence that LSA3 is involved in antigenic variation.

Line 331 In reference 18 the cut-off of 50 plasmid copies was established for a Ct of 38 or less.

A correction to the naming of the two species has been recently made (Snounou et al. Trends in Parasitology 2004 40:21), and it would be suitable to adopt the terms P. ovalecurtisi and P. ovalewallikeri in a revised manuscript.

**Summary and General Comments**

Reviewer #1: In this simple report, He and colleagues describe an RTqPCR assay for the detection and differentiation of Plasmodium ovalecurtisi and P. ovalewallikeri. They argue that their assay is ‘less complex’ and ‘streamlined’ compared to other assays, and advocate its use in field studies. 

Major points.

Given that their assay is of equal or less sensitivity than other previously reported assays, it is difficult to see the circumstances in which it may replace those currently in use. This assay can only be used in well-funded laboratories with access to real time PCR facilities, and such laboratories would presumably fair better using the more sensitive current methodologies. Of particular concern is the fact that their P. ovalewallikeri assay is ten times less sensitive than their P. ovalecurtisi assay, rendering it, in my opinion, useless in its ability to assay the relative prevalence of the two species, or, indeed, to identify P. ovalecurtisi in relatively low parasite density infections. The authors state in their discussion that their ‘assays sensitivity at lower parasite densities could be further improved’ (line 381). If it can be further improved, then it is perhaps better to do so before publishing and advocating its use. 

My second major concern is that the assay was developed using only sequences from a limited region of central Africa. It is possible, therefore, that it may not be suitable for use outside that region, especially the P. ovalecurtisi assay, which targets a non-coding region (and so is potentially polymorphic). It would be nice to see some attempt to address this, either just through analysis of polymorphism of the targets in a more diverse pool of P. ovale parasites, or through wet lab testing of isolates from outside central Africa. 

Other points

Nomenclature. Throughout the manuscript, the two ovale species are referred to either as “P. ovale curtisi” and ”P. ovale wallikeri” or as “Poc” and “Pow”. It has recently been suggested that the names “P. ovalecurtisi” and “P. ovalewallikeri” are preferable, as binomials reflect the species nature of the two parasites. I see no need to abbreviate to ‘Poc’ and ‘Pow’ (we don’t call P. falciparum “P. fal”, or P. malariae “P. mal”). I suggest correcting to “P. ovalecurtisi” and “P. ovalewallikeri”. 

Line 25. ‘Species’ are, by definition, ‘non-recombining’. There is no need to add this phrase before the word ‘species’. 

Line 59. The evidence for this isn’t convincing at the moment, and is somewhat contradictory. 

Line 71. I’m not sure ‘most malaria programes’ prioritise P. falciparum… which programmes are these? Elimination programmes? Control programmes? P. vivax is the major issue outside Africa. Perhaps it’s best to specify ‘malaria control programmes within Africa’? 

Line 77 the phrase ‘distinct non-recombing species’ can be reduced to just ‘species’. 

Line 80. There are no single step assays, surely? 

Line 83. What do the authors mean by ‘conventional malaria diagnostic assays relying on microscopy’. Do they mean simply ‘microscopy’?

Line 98. ‘this assay requires multiple steps’. What does this mean? ALL assays require multiple steps… And a ‘long turnaround time’? Perhaps it would be helpful if a table is included comparing the various assays for detection and discrimination of the two species. The table could include the sensitivities, the costs, the number of ‘steps’ etc. It would be nice to include the methodology of Nundu et al in reference 47, who performed a simple nested PCR then sequenced the P. ovale positives. 

Line 105. Perhaps the authors could explain here what a ‘duplex qualitative real-time PCR assay’ is for the general reader. 

Line 129 What does the term ‘well-characterised’ mean here?

Line 152. What constitutes a ‘replicate’ in this context?

Line 154 What does the term ‘well-characterised’ mean here?

Line 187. Does this mean that of the 64 previously P. ovale positive samples, only 44 were found to be positive on repeated analysis? This is quite a significant result – could the authors speculate as to the reason of this significant discrepancy?

Line 232. It is specified that 3 uL of DNA template was used in the assay. Could the authors explain how much total blood volume this is equivalent to? How much blood was used for DNA extraction? Without this data, it is impossible to assess the actual sensitivity of the assay. 

Line 239, as above. ‘equivalents/uL’… uL of what? Blood or DNA solution? The approximate volume of blood assayed in each reaction needs to be given. 

Line 242. What was the negative control for these assays? Was it uninfected blood extracted in the same way as the samples?

Line 255 is rather an odd sentence. It is in effect saying that sensitivity was excellent when there was lots of parasite DNA present. That’s not a good measure of ‘sensitivity’. 

Line 258: the number of parasites per ‘uL’ is given again – it is essential to know whether this is uL of DNA solution or of original blood. 

Line 259 – the assays sensitivity of 80% is compared to what? Blood smear positive? Previous qPCR results? 

Line 267. The 64 P. ovale spp positive samples had been determined as positive using a previous assay, I think. The authors could only confirm 44 of these to be truly P. ovale positive using the same assay… if I’ve got this right, then the epidemiological analysis should only be performed on the 44 confirmed P. ovale ceases, surely? 

Line 267 onwards. The epidemiology section seems somewhat shoe-horned into the paper, and doesn’t really fit with the rest of it – as the risk factor calculations were not split into P. ovalewallikeri and P. ovalecurtisi, this analysis does not depend on the new assay under discussion, and is based on a previously carried out analysis, it seems superfluous to the paper as a whole. It would be much better to just give the breakdown of the two species compositions in the original sample set. Related to this, I fail to see the significance of the performance of an ‘inverted probability weighting analysis’ to extrapolate from 37 samples to 44 (?). 

Line 301. The authors describe their assay as “highly specific’. Yet, the P. ovalecurtisi component is 10 times more sensitive than the Pow component. I can’t reconcile these two statements. They state it can be used to ‘improve our understanding of their epidemiology in malaria-endemic countries’ yet it is less sensitive than other RTqPCR assays and conventional nested PCR? How will it, therefore, improve our understanding?

Line 305 “distinct, non-recombining species” This suggests that there are indistinct, recombining species in existence…. 

Line 307. Why are higher volumes of DNA required in other assays? The author’s assay is less sensitive, so would require a larger volume of DNA to reach the same level of sensitivity. 

Line 308. ‘multiple steps’ is vague. Again, a table comparing the methods alluded to with the authors’ assay would be useful. 

Line 319. In world terms, the samples weren’t collected from particularly ‘geographically diverse sites’, all of them being in central Africa. 

Line 325. Could the ‘advantageous performance characteristics’ be given here?

Line 345. How were these prevalences determined? 

Line 354 “Existing evidence indicates that the most prevalent P. ovale spp. vary across different countries’. Could the authors rewrite this sentence for clarity…?

Line 356. What does ‘Poc more prevalent in symptomatic individuals’ mean? More prevalent than in asymptomatic individuals? More prevalent than Pow in symptomatic individuals? If the latter, then this will always appear to be the case when performing the authors' assay, as the Poc sensitivity is 10 times that of Pow.

Reviewer #2: See attachment

Reviewer #3: The authors have mined the genome of the two P. ovale species and selected in each a segment of DNA that is repeated in the genome and designed sets of oligonucleotide primers and corresponding probes that were then tested in real-time PCR assays for specificity and sensitivity. One set for each species that was identified as giving the optimal results was then tested using a selected set of P. ovale-positive samples (identified in a previous study), and this new protocol is advocated to perform better/to be more practical than others and therefore to serve for high throughput surveys in endemic countries. The authors then derived some epidemiological conclusions from the data obtained from these samples.

Major Comments

1) Whereas the approach of the authors (targeting internally repeated short segments) is original for P. ovale and the methodology and data are sound, the main thrust of this manuscript is that their methodology (duplex real-time PCR) is better suited than all other published methods to date for investigations on the epidemiology/biology of these parasite species, with the main supporting argument is that it is less time-consuming (fewer steps between sample collection and assay results). However, many factors (some quoted by the authors and others not) trump this minor advantage. First, the ten-fold lower sensitivity of detection for one species (Pow) as compared to the other (Poc) is a major limitation that would bias data from all field surveys, because a) it is well-known that P. ovale infections have low parasite densities past the primary peak that often persist for long durations, and b) low parasite densities, especially in mixed and asymptomatic infections, are only detected when highly sensitive molecular assays capable of detecting very low parasite levels (1 P/µl or less).

2) It is difficult from the description given to work out the actual sensitivity that was obtained. The values (4,2 and 41,2 P/µL for Poc and Pow, respectively) quoted were from the Probit analysis using diluted plasmids. There is no indication as to the Plasmodium sequence that these plasmids contained. Moreover, it is not clear to what volume of blood the 3 µL DNA template obtained from Chelex-purified filter papers corresponds. Assuming 10 µL per filter spot, and 100 µL of DNA solution post-Chelex extraction, the 3 µL would correspond to 0,3 µL of blood. Is this the case? Furthermore, were the DNA templates used obtained from leukodepleted blood?

3) One final point concerns the potential loss of sensitivity when the reaction is multiplexed, which is the case here. The authors should conduct experiments in which different proportions of DNA template from each species are mixed to assess whether the assay is capable of detecting a population of Poc or Pow when present as a minor proportion of the parasites (for instance in samples with high P. falciparum parasitaemias and only a few P. ovale per µL of blood).

4) There is little known concerning the global diversity of the repeats in the putative lsa3 gene and for the untranslated repeats on which the assay is based. Although it is likely that potential variations would not affect sensitivity, all the samples used to validate the assay were collected from the DRC and a few from Tanzania. It would have been very useful to include samples from West Africa and Oceania.

Ultimately the authors have not provided sufficient evidence to support their claim that this novel method is superior to others for field investigations of the two P. ovale species. One might argue that the high rate of false negatives is a major disadvantage of the assay: 11 of the 55 P. ovale 5 samples used to validate the assay were not identified, i.e. a 20% failure rate, even if this was mainly in cases with low parasite burdens, A shorter processing time does not compensate for the lower sensitivity, and the techniques used still require a well-equipped laboratory.

5) The conclusions from the epidemiological analyses are at best speculative because they are based on a restricted number of samples from symptomatic cases collected from various locations in the DRC, and because of the bias in the detection of Pow. The authors acknowledge the limitations in the Discussion. I suggest that this section is omitted from the manuscript.

PLOS authors have the option to publish the peer review history of their article (what does this mean?). If published, this will include your full peer review and any attached files.

Reviewer #1: Yes: Richard Culleton

Reviewer #2: No

Reviewer #3: No
---

## [Decision Letter · Decision Letter 1]

15 Aug 2024

Dear Dr. Parr,

We are pleased to inform you that your manuscript 'Development of new real-time PCR assays for detection and species differentiation of Plasmodium ovale' has been provisionally accepted for publication in PLOS Neglected Tropical Diseases.

Best regards,

Georges Snounou, Ph.D.

Guest Editor

Paul Mireji

Section Editor

The decisions of the reviewers are contradictory. However, I will consider that the assay you present has some value, if of limited use because of the poor sensitivity to one of the P. ovale species, and I am therefore recommending acceptance of the manuscript.

I strongly recommend that you adopt the correct nomenclature for the two species, namely P. ovalecurtisi and P. ovalewallikeri. Colin Sutherland and I are in agreement that the trinomial is misleading and contravenes the ICZN rules (as this particular nomenclature is restricted to sub-species).

With best regards and apologies for the delay in posting my recommendation.

Georges

Reviewer's Responses to Questions

**Key Review Criteria Required for Acceptance?**

**Methods**

-Are the objectives of the study clearly articulated with a clear testable hypothesis stated?

-Is the study design appropriate to address the stated objectives?

-Is the population clearly described and appropriate for the hypothesis being tested?

-Is the sample size sufficient to ensure adequate power to address the hypothesis being tested?

-Were correct statistical analysis used to support conclusions?

-Are there concerns about ethical or regulatory requirements being met?

Reviewer #1: (No Response)

Reviewer #2: See comments below

**Results**

-Does the analysis presented match the analysis plan?

-Are the results clearly and completely presented?

-Are the figures (Tables, Images) of sufficient quality for clarity?

Reviewer #1: (No Response)

Reviewer #2: See comments below

**Conclusions**

-Are the conclusions supported by the data presented?

-Are the limitations of analysis clearly described?

-Do the authors discuss how these data can be helpful to advance our understanding of the topic under study?

-Is public health relevance addressed?

Reviewer #1: (No Response)

Reviewer #2: See comments below

**Editorial and Data Presentation Modifications?**

Reviewer #1: (No Response)

Reviewer #2: See comments below

**Summary and General Comments**

Reviewer #1: The authors have made some modifications to their manuscript, toning down the claims of the previous version and removing the epidemiological survey component. Whilst the manuscript itself is somewhat improved, the assay still remains sub-optimal, in that the sensitivities of the two components are hugely different, and the merit of the assay versus currently used methodologies is not apparent. It is a shame that the authors have not revisited their assay in an attempt to increase the sensitivity of the P. ovalewallikeri assay, as this would make an important improvement to the work. This methodology leading to the design of this assay has potential, and it would be good to see that realised. As it stands, it feels as if the scope (and, indeed, necessity) for assay improvement renders the current manuscript more of a description of a work in progress rather than a fully optimised and practical assay.

Specific Points

Sequence Diversity

In my previous review I raised the concern that only African P. ovale spp. were used in the design and testing of this assay. In answer to this concern, the authors responded with reference to the low level of variability in the target sequences for samples from the DRC, Tanzania, Cameroon and Ethiopia. Without labouring the point, P. ovale is endemic outside Africa. If it is difficult to assess the level of diversity in non-African samples, then perhaps it should be made clear in the manuscript that this assay is optomised for African isolates, and may not be suitable for use elsewhere.

Table S1

The usefulness of this table is somewhat reduced by the omission of data for the authors’ own assay. In order to compare the assay to those previously published (the point of the previous request to include a table), it is necessary to include this data.

Table S4 and S5

These tables show the sensitivities of the assays using plasmid DNA. The ‘parasite density’ column is given as ‘parasites/ul’ (sic); I’m sorry if this is explained in the text, and I’ve missed it, but how are the authors extrapolating from ‘copy number (of target)’ to ‘parasite density’? Does each plasmid contain the same copy number of the target as a parasite genome? If not (and there are more copies per genome), then the ‘parasite density’ should be adjusted accordingly. This should be explained in a footnote to the table. Alternatively, the second column should be labelled ‘plasmid DNA copy number’. For example, there are nine copies of the P. ovalecurtisi target per parasite genome, but only one copy per plasmid. Does this mean that the “parasite density (parasites/ul)” has been multiplied nine times from the plasmid copy number sensitivity? So, for Poc, the assay is reliable only down to 10 ‘parasites/ul’; does this equate to 900 copies of the plasmid? It would be useful to clarify this.

Line 106. While differentiation of the two species is not possible, detection certainly is. Please modify sentence for clarity. Perhaps just “Differentiation of P. ovalecurtisi and P. ovalewallikeri is not possible using microscopy” is a more accurate sentence.

Nomenclature

The unanimous consensus of the malaria research community is that the trinomial names “P. ovale curtisi” and “P. ovale wallikeri”, are incorrect, confusing, and constitute ‘nomina nuda’. Snounou et al (2023a), offered a simple and positively received solution to the naming issue, to which only one objection was raised (Slapeta, 2023, who also acknowledge the incorrect status of the trinomials). This objection was subsequently shown to be erroneous and unsound (Snounou et al 2023b). In the interests of clarity, consensus and consistency in the literature pertaining to these two parasites, the use of the binomials as designated by Snounou et al (2023) is strongly advised.

Reviewer #2: Dear editor,

Please find below, my review of the revision of the manuscript PNTD-D-23-01354-R1.

The manuscript has globally improved in its presentation following the corrections and modifications made by the authors according to the reviewer's comments. i am glad to see the 'so called' epidemiological section out.

Regarding the assay itself, as share before, its the development and setting up are correct and sounded following expected steps.

Although, as new P. ovale curtisi and P. ovale wallikeri genomes have been released recently (Higgins et al 2024), it is surprising that the authors did not attempt to verify if their assay is still functional /compatible on these new genomes and also check for possible other repetitive elements to try improve the main weakness of their assay which is a 10 fold variations between the limits of detection of the two P. ovale species.

at the end of the day, the question of the usefulness of this assay remains, as it is of strictly no interest for the medical doctors and unlikely to be adopted in resources limited settings laboratories where the potra assay a been adopted. However, this assay might still find its place in laboratories covering national programs and screening large number of samples where it could be incorporated and combined within establish workflows.

PLOS authors have the option to publish the peer review history of their article (what does this mean?). If published, this will include your full peer review and any attached files.

Reviewer #1: No

Reviewer #2: No

---

## [Editor Report · Acceptance letter]

3 Sep 2024

Dear Dr. Parr,

We are delighted to inform you that your manuscript, "Development of new real-time PCR assays for detection and species differentiation of Plasmodium ovale," has been formally accepted for publication in PLOS Neglected Tropical Diseases.

Best regards,

Shaden Kamhawi

co-Editor-in-Chief

Paul Brindley

co-Editor-in-Chief
